# What message appeal and messenger are most persuasive for COVID-19 vaccine uptake: Results from a 5-country survey in India, Indonesia, Kenya, Nigeria, and Ukraine

**Rupali J. Limaye** [1,2,3,4] *, **Kristian Balgobin** [1,2], **Alexandra Michel** [1,2], **Gretchen Schulz**[5], **Daniel J. Erchick**[1]

1 Department of International Health, Johns Hopkins Bloomberg School of Public Health, Baltimore, Maryland, United States of America, 2 International Vaccine Access Center, Johns Hopkins Bloomberg School of Public Health, Baltimore, Maryland, United States of America, 3 Department of Epidemiology, Johns Hopkins Bloomberg School of Public Health, Baltimore, Maryland, United States of America, 4 Department of Health, Behavior & Society, Johns Hopkins Bloomberg School of Public Health, Baltimore, Maryland, United States of America, 5 Department of Population, Family and Reproductive Health, Johns Hopkins Bloomberg School of Public Health, Baltimore, Maryland, United States of America

\* rlimaye@jhu.edu

**Data Availability Statement:** All relevant data are within the paper and its Supporting information files.

## Abstract

Effective strategies to encourage COVID-19 vaccination should consider how health communication can be tailored to specific contexts. Our study aimed to evaluate the influence of three specific messaging appeals from two kinds of messengers on COVID-19 vaccine acceptance in diverse countries. We surveyed 953 online participants in five countries (India, Indonesia, Kenya, Nigeria, and Ukraine). We assessed participants' perceptions of three messaging appeals of vaccination—COVID-19 disease health outcomes, social norms related to COVID-19 vaccination, and economic impact of COVID-19—from two messengers, healthcare providers (HCP), and peers. We examined participants' ad preference and vaccine hesitancy using multivariable multinomial logistic regression. Participants expressed a high level of approval for all the ads. The healthcare outcome–healthcare provider ad was most preferred among participants from India, Indonesia, Nigeria, and Ukraine. Participants in Kenya reported a preference for the health outcome–peer ad. The majority of participants in each country expressed high levels of vaccine hesitancy. However, in a final logistic regression model participant characteristics were not significantly related to vaccine hesitancy. These findings suggest that appeals related to health outcomes, economic benefit, and social norms are all acceptable to diverse general populations, while specific audience segments (i.e., mothers, younger adults, etc.) may have preferences for specific appeals over others. Tailored approaches, or approaches that are developed with the target audience's concerns and preferences in mind, will be more effective than broad-based or mass appeals.

**Funding:** This work was supported by the Sabin Vaccine Institute (https://www.sabin.org/) under grant 050119-00 to RL. The funders had no role in study design, data collection and analysis, decision to publish, or preparation of the manuscript.

**Competing interests:** The authors have declared that no competing interests exist.

## Introduction

Vaccine hesitancy, the reluctance or refusal to vaccinate despite availability of vaccines, has impacted efforts to control the SARS-CoV-2 (COVID-19) pandemic around the world. COVID-19 vaccine behavior, as with other vaccines, is complex, context-specific, and dependent upon a multitude of influences [1, 2]. Disinformation and misinformation during the COVID-19 pandemic have spread through social media platforms and have significantly impacted vaccine confidence and uptake [3]. Hesitancy toward vaccination has been linked to outbreaks of vaccine preventable disease; and with the COVID-19 pandemic, evidence indicates that unvaccinated individuals are at much higher risk of severe outcomes [4].

The COVID-19 vaccine in particular faces significant hesitancy and low uptake globally. As of March 2022, approximately 65% of the global population have received at least one dose of the COVID-19 vaccine [5]. A recent review found that low rates of COVID-19 vaccine uptake are most prominent within the Middle East, Russia, Africa and several European countries [6]. COVID-19 vaccine hesitancy varies by country and context [2].

India has faced some of the most difficult COVID-19 surges, with more than 521,000 deaths [7], making vaccination uptake essential to reduce morbidity and mortality. A recent survey found that about 37% of respondents were unsure or would not obtain the vaccine, and this is likely an underestimate of hesitancy as most of these respondents were urban and had higher levels of educational attainment [8]. In Indonesia, vaccine acceptance was anticipated to be higher than 90% based on recent surveys, however, currently only 58% of the population have been fully vaccinated [6, 9]. Kenya faces significant COVID-19 vaccine hesitancy, with approximately 37% of people not intending to be vaccinate due to low perceived risk, vaccine efficacy, and other cultural/ religious reasons [10]. A more stark picture can be seen in Nigeria, where early attitudes toward the vaccine were somewhat positive, with 65% of the population planning to vaccinate, while current vaccination rates struggle to break 5% [11, 12]. Lastly, with a long history of vaccine hesitancy, Ukraine faces significant challenges in hesitancy related to vaccine safety and efficacy [6, 13] and will likely face increasing challenges in uptake due to on-going military conflict.

While the contexts of these countries vary drastically, there are common threads of why so many people are hesitant to accept COVID-19 vaccines, including: distrust in government and health authorities, concerns regarding vaccine safety and efficacy, and false information about effects of the vaccine [3, 6]. Although the reasons behind vaccine hesitancy vary, the main contributor to vaccine uptake across all countries is one's interest in personal protection against COVID-19 [2]. Thus, vaccine communication should be tailored based on context and focus on safety and efficacy, advantages of vaccination, and the social norms associated with vaccine uptake particularly when countries are experiencing an increase in COVID-19 cases [2, 3].

Many communication approaches have been proposed to address vaccine hesitancy. Broadly, approaches include dialogue-based, reminder/recall, and multi-component approaches, among others [1, 14]. A review focused on strategies to mitigate vaccine hesitancy found that communication interventions aimed at reducing vaccine hesitancy are most effective when a diverse range of message appeals, approaches, and messengers are used [14]. Most interventions to reduce vaccine hesitancy have been conducted in high-income settings, have focused on a few vaccines (e.g., HPV, influenza, and MMR) and examine multi-component interventions–with varied combinations of appeal, approach, and messenger strategies [15].

Health communication strategies often utilize a singular appeal to attract a recipient's attention, and an appeal serves as a guide for what to focus on in a message. A recent review found that interventions using messages tailored to behavior change, personal narratives, and peer approaches that utilize cultural and societal norms were effective approaches to address

vaccine concerns and increase acceptance [15]. Beyond the appeal, the messenger can also strongly impact the effectiveness of the message. Prior evidence has highlighted the importance of a healthcare professional vaccine recommendation as one of the strongest drivers of vaccine uptake [2].

Effective strategies to encourage COVID-19 vaccination uptake must consider how different aspects of health communication can be tailored to specific contexts. This gap in research informed our study aimed to evaluate the influence of three specific messaging appeal framings—COVID-19 disease health outcomes, social norms related to vaccination, and economic impact of COVID-19—of vaccination from two kinds of messengers, healthcare providers and peers, on COVID-19 vaccine acceptance in a diverse set of countries.

## Methods

### Ethics statement

The study protocol was approved by the Johns Hopkins Bloomberg School of Public Health Institutional Review Board (No:00018565) prior to study initiation. A brief consent statement appeared on the screen used for the survey. Informed consent was obtained online through the presentation of the consent message and asking a single question about whether the participant would like to participate in the study.

### Participants

We conducted online surveys in five countries (India, Indonesia, Kenya, Nigeria, and Ukraine), using SurveyMonkey, a survey and panel research company. We chose these countries given their history with vaccine hesitancy, as well as related COVID-19 morbidity and mortality. For data quality checking for online panel participants, SurveyMonkey employs bot and fraud detection when recruiting panel participants [16]. For inclusion in our online panels, participants had to be located in the target country, over the age 18, and proficient in written English.

### Procedures

As this was a descriptive and exploratory study, we did not have apriori hypotheses that we sought to test. We were interested in ad preferences among users. A total of 1,894 responses were collected. Gender balance was a parameter we applied to achieve approximately 50% representation of both males and females. To ensure high-quality results for analysis, we excluded responses from the analysis if there were indications of poor attention or "speed racing" responses. Exclusion criteria included unrealistically short completion times (answering the 55 survey questions in less than 5 minutes, n = 704), failure of attention checks (n = 217), failure to answer all questions, including open text responses (n = 20). Of the total 1,894 survey responses collected, 953 met our quality criteria for inclusion in the analysis (Fig 1). We undertook these quality checks to ensure that we were able to capture the highest quality responses. We determined our sample size based upon calculation of confidence interval width around an expected sample proportion of vaccine hesitancy across what is known about vaccine hesitancy in each country we included. We estimated vaccine hesitancy prevalence to range between 10–50%; a sample size of 500 participants will yield two-sided 95% confidence intervals with a width of approximately 0.05.

The survey included 55 questions. It included socio-demographic questions, including age (18–24, 25–39, 40–64, 65+), gender (male, female, other, prefer not to say), level of education (secondary or high school, bachelor's degree or 4-year college degree, graduate level degree, or

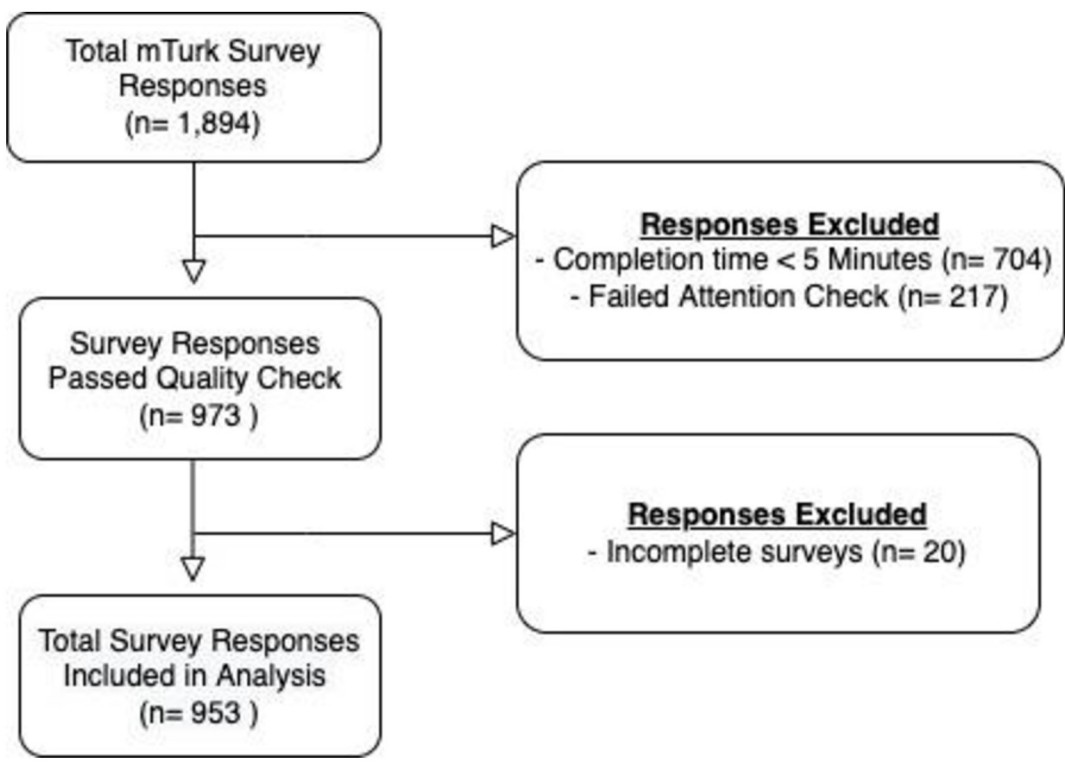

**Fig 1. Participant flowchart.**

other), and pregnancy status (yes, no, not-applicable). The survey was pretested with 10 SurveyMonkey users.

Each participant viewed six ads, which were broadly composed of two elements: a distinct messenger and a distinct appeal. The messengers included a healthcare provider image, which depicted a medical provider talking to a patient, and a peer image, which depicted two people speaking to each other. We included the following appeals: health outcome, which focused on the risk of COVID-19 disease and the protective effect of vaccination against disease; economic benefit, which focused on loss of work time and income due to COVID-19 infection and the protective effect of vaccination against economic loss; and social norms, which focused on how most people have received the COVID-19 vaccine and the protective effect of vaccination for the community. We chose these appeals after conducting a scoping review to determine which appeals may be most effective in nudging an individual to accept a vaccine.

After each ad, six questions covering participant level of agreement specific to each ad were asked, including relevance (*Please indicate your level of agreement with the following statement*: *this ad was relevant to me*), motivation to get the COVID-19 vaccine (*Please indicate your level of agreement with the following statement*: *this ad motivates me to get the COVID-19 vaccine*), motivation to get COVID-19 vaccination for their child (if applicable) (*Please indicate your level of agreement with the following statement*: *this ad motivates me to get the COVID-19 vaccine for my child under 18 years of age*), if the ad was designed for someone like the participant (*Please indicate your level of agreement with the following statement*: *this ad was designed for people like me*), and if the ad would prompt the participant to tell someone about the COVID-19 vaccine (*Please indicate your level of agreement with the following statement*: *this ad would prompt me to tell someone about the COVID-19 vaccine*). Participants were also asked to

indicate which ad of the six would motivate them the most to get the COVID-19 vaccine (*Which ad motivates you most to get the COVID-19 vaccine*?).

Three questions were used to assess participant vaccine hesitancy. Participants were asked if they had ever delayed getting a recommended vaccine through a yes/no response (*Have you ever delayed getting a recommended vaccine or decided not to get a recommended vaccine for reasons other than illness or allergy*?*)*. Two questions asked participants about their level of agreement related to COVID-19 vaccine safety concern (*How concerned are you that a COVID-19 vaccine might not be safe*?*)*, and participant perception of vaccine effectiveness (*How concerned are you that a COVID-19 vaccine might not prevent the disease*?*)*.

To further examine the reason why an individual may not want to receive the COVID-19 vaccine, participants were asked to rank six statements based on level of concern, with one being the most concerning, to six being the least concerning (*There are multiple reasons why someone may not want to get the COVID-19 vaccine. Please rank these reasons in order, with 1 being the most concerning, to 6 being the least concerning.*). Concerns included safety (*I do not feel the vaccine is safe*), vaccine effectiveness (*I do not feel the vaccine is effective*), trust in government (*I do not trust the government*), vaccine experience (*People I know have not had a good experience getting the vaccine*), and cost (*It is cost prohibitive for me to get the vaccine*), and belief in the existence of COVID (*I believe COVID-19 is not real*).

We constructed an ordinal scale from 0 to 3 to describe participants' level of vaccine hesitancy by assigning 1 point for each of three yes/no questions. We then examined the distribution of scores and established a cut-off to stratify participants into two groups (0–1 and 2–3), which we defined as lower hesitancy and higher hesitancy. Statistical analysis was performed in Stata 16.1 (Stata Corp, College Station, TX). The study received ethical approval from the Institutional Review Board at the *(blinded for review)*.

## Statistical analysis

We summarized participant characteristics overall, participant responses and assessed differences between countries using chi-squared tests. We collapsed ad preference responses into binary variables (strongly agree/agree vs. strongly disagree/disagree) and examined the proportion responding positively across constructs for each ad. Variables for some participant characteristics were collapsed due to small numbers in some categories, including age (collapsed to <40, ≥40 levels for regression models), gender (female and male levels used for regression models as the "Prefer not to say" option had only n = 3 observations).

We examined ad preference by asking participants which ad they most preferred and comparing responses by country and participant characteristics using multivariable multinomial logistic regression. Individual country models were used to assess within-country differences. Participants were pooled due to the identical components across the five countries, such as inclusion and exclusion criteria, recruitment methods, measures, intervention approach, and procedures of study implementation [17]. To explore further differences (India, Indonesia, Kenya, Nigeria and Ukraine), a pooled participant model was used to assess cross-country preferences. We estimated relative risks and 95% confidence intervals using ad 1 as the reference group (*Health Outcome–Healthcare Provider* ad) and a binary variable for vaccine hesitancy as our primary characteristic of interest. To examine associations for ads depicting healthcare providers and peers independently, we stratified on this condition and modeled the relationships in separate multivariable multinomial models. Included in models were participant characteristic data known from the literature to be associated with vaccine attitudes, including age and gender.

## Results

A total of 1,894 participants responded to the survey. We excluded participants using the following data quality checks: survey completion <5 minutes (n = 704, 37.2%) and failed attention check (n = 217, 11.5%). We also excluded participants for incomplete survey responses (n = 20, 1.1%). Table 1 describes the 935 participants included in our analysis. The majority of participants were 18–24 years old (n = 463, 48.6%), female (n = 487, 51.3%), and had a bachelor's degree (n = 471, 53.3%). Most participants reported higher vaccine hesitancy (n = 698, 73.3%); however, more than two-thirds of the participants reported they were vaccinated against COVID-19 (n = 740, 79.6%).

More than a quarter of participants (n = 288, 31.8%) reported having ever delayed or refused a recommended vaccination. Most participants were at least slightly concerned that the vaccine might not prevent COVID-19 disease (n = 512, 53.7%) or might not be safe (n = 595, 62.43%). The majority of participants reported more concerns (participants were either moderately or extremely concerned), when asked about the vaccine for pregnant women (n = 531, 55.7%) and children under the age of 18 (n = 540, 56.7%). Few participants had a score of 0 on our vaccine hesitancy scale (n = 77, 8.1%). The vaccine hesitancy scale distribution across the three questions for the remaining participants was 1 (n = 178, 18.7%),

**Table 1. Participant characteristics (n = 935).**

| Characteristic* | No. (%) | | | | |
|---|---|---|---|---|---|
| Country | India (n = 207) | Indonesia (n = 232) | Kenya (n = 194) | Nigeria (n = 152) | Ukraine (n = 168) |
| *Age* | | | | | |
| 18–24 | 115 (55.6) | 128 (55.2) | 111 (57.2) | 81 (53.3) | 28 (16.7) |
| 24–39 | 79 (38.7) | 93 (40.1) | 71 (36.6) | 49 (32.2) | 91 (54.2) |
| 40–64 | 13 (6.3) | 11 (4.7) | 9 (4.6) | 16 (10.5) | 45 (26.8) |
| 65+ | 0 (0.0) | 0 (0.0) | 3 (1.6) | 6 (4.0) | 4 (2.4) |
| *Gender* | | | | | |
| Female | 106 (51.2) | 128 (55.2) | 95 (49.2) | 67 (44.1) | 91 (54.8) |
| Male | 101 (48.8) | 104 (44.8) | 98 (50.8) | 85 (55.9) | 75 (45.2) |
| *Education* | | | | | |
| Secondary | 19 (9.6) | 96 (42.5) | 29 (17.1) | 24 (15.9) | 28 (18.0) |
| Bachelor's degree | 97 (49.0) | 111 (49.1) | 119 (70.0) | 86 (57.0) | 58 (37.2) |
| Graduate degree | 82 (41.4) | 19 (8.4) | 22 (12.9) | 41 (27.2) | 70 (44.9) |
| *Pregnant+* | | | | | |
| No | 85 (81.7) | 105 (83.3) | 79 (84.0) | 57 (86.4) | 62 (72.1) |
| Yes | 19 (18.3) | 21 (16.7) | 15 (16.0) | 9 (13.6) | 24 (27.9) |
| *COVID-19 Vaccinated* | | | | | |
| No | 12 (5.9) | 8 (3.5) | 49 (25.5) | 70 (47.0) | 51 (32.5) |
| Yes | 192 (94.1) | 220 (96.5) | 143 (74.5) | 79 (53.0) | 106 (67.5) |
| *Vaccine Hesitancy* | | | | | |
| Lower Hesitancy | 89 (43.0) | 79 (34.1) | 40 (20.6) | 17 (11.9) | 30 (17.9) |
| Higher Hesitancy | 118 (57.0) | 153 (66.0) | 154 (79.4) | 135 (88.8) | 138 (82.1) |

Participant demographic characteristics across the five countries.

a* Of 935 total observations, missingness for each variable was as follows: age (n = 0, 0%), gender (n = 3, 0.3%), education (n = 52, 5.6%), pregnant (n = 477, 51.0%), COVID-19 vaccinated (n = 23, 2.5%), vaccine hesitancy (n = 0, 0%).

+ Pregnancy status assessed among participants identifying as women.

2 (n = 462, 48.5%), and 3 (n = 236, 24.8%). We categorized participants as lower hesitancy (0 or 1 concerns) (n = 255, 26.8%) and higher hesitancy (2 or 3 concerns) (n = 698, 73.2%).

The majority of participants indicated higher vaccine hesitancy in every country: India (57.0%), Indonesia (66.0%), Kenya (79.4%), Nigeria (88.8%), and Ukraine (82.1%). However, vaccine hesitancy and participant characteristics were not significant in a final logistic regression model. Vaccine hesitancy regression results illustrated two borderline associations with higher vaccine hesitancy. Kenyan participants with bachelor's degrees (OR = 2.42, p = .059) and older (40+ or older) Nigerian participants (OR = 0.23, p = .059) were more likely to be vaccine hesitant. Furthermore, using logistic regression to compare lower vaccine-hesitant participants to the higher vaccine-hesitant participants, using India as a reference group, participants from Kenya (OR = 2.90, P< .01, 95% CI [1.86, 4.52]), Nigeria (OR = 6.31, P< .01, 95% CI [3.54, 11.26]), and the Ukraine (OR = 4.09, P< .01, 95% CI [2.43, 6.88]), were more likely to have higher vaccine hesitancy than participants from India.

In general, participants agreed with the message (level of agreement > 90%) across all six ads (Table 2). For all countries except Kenya, the majority of participants indicated a preference for the health outcome–healthcare provider ad. Participants in India (26.6%), Indonesia (31.9%), Nigeria (34.9%), and Ukraine (31.6%) reported the health outcome–healthcare provider ad as the ad most likely to motivate them to get the COVID-19 vaccine (Fig 2). Participants in Kenya reported a preference for the health outcome–peer ad (35.1%) followed by the health outcome–healthcare provider ad (28.9%). Across country pooled descriptive analysis

**Table 2. Participant preferences for message aspects across six ads (n = 935).**

| | No. (%) | | | | | |
|---|---|---|---|---|---|---|
| | Health Outcome Healthcare provider | Health Outcome Peer | Economic Healthcare provider | Economic Peer | Social norm Healthcare provider | Social norm Peer |
| **I agree with the message in the ad** | | | | | | |
| Strongly Agree/Agree | 890 (93.4) | 892 (93.6) | 872 (91.5) | 882 (92.6) | 874 (91.7) | 873 (91.6) |
| Strongly Disagree/ Disagree | 63 (6.6) | 61 (6.4) | 81 (8.5) | 71 (7.5) | 79 (8.3) | 80 (8.4) |
| **Ad would prompt me to tell someone about the COVID-19 vaccination** | | | | | | |
| Strongly Agree/Agree | 857 (89.9) | 880 (92.3) | 852 (89.4) | 856 (89.8) | 884 (92.8) | 862 (90.5) |
| Strongly Disagree/ Disagree | 96 (10.1) | 73 (7.7) | 101 (10.6) | 97 (10.2) | 69 (7.2) | 91 (9.6) |
| **Ad was designed for people like me** | | | | | | |
| Strongly Agree/Agree | 805 (84.5) | 833 (87.4) | 807 (84.7) | 812 (85.2) | 856 (89.8) | 825 (86.6) |
| Strongly Disagree/ Disagree | 148 (15.5) | 120 (12.6) | 146 (15.3) | 141 (14.8) | 97 (10.2) | 128 (13.4) |
| **As was relevant to me** | | | | | | |
| Strongly Agree/Agree | 880 (92.3) | 862 (90.5) | 845 (88.7) | 840 (88.1) | 874 (91.7) | 857 (89.9) |
| Strongly Disagree/ Disagree | 73 (7.7) | 91 (9.6) | 108 (11.3) | 113 (11.9) | 79 (8.3) | 96 (10.1) |
| **Ad motivates me to get my child the COVID-19 vaccine** | | | | | | |
| Strongly Agree/Agree | 643 (67.5) | 612 (64.2) | 584 (61.3) | 571 (59.9) | 587 (61.6) | 587 (61.6) |
| Strongly Disagree/ Disagree | 156 (16.4) | 152 (16.0) | 173 (18.2) | 170 (17.8) | 152 (16.0) | 161 (16.9) |
| Not a parent | 154 (16.2) | 189 (19.8) | 196 (20.1) | 212 (22.3) | 214 (22.5) | 205 (21.5) |
| **Ad motivates me to get the COVID-19 vaccine** | | | | | | |
| Strongly Agree/Agree | 876 (91.9) | 870 (91.3) | 867 (91.0) | 843 (88.5) | 860 (90.2) | 847 (88.9) |
| Strongly Disagree/ Disagree | 77 (8.1) | 83 (8.7) | 86 (9.0) | 110 (11.5) | 93 (9.8) | 106 (11.1) |

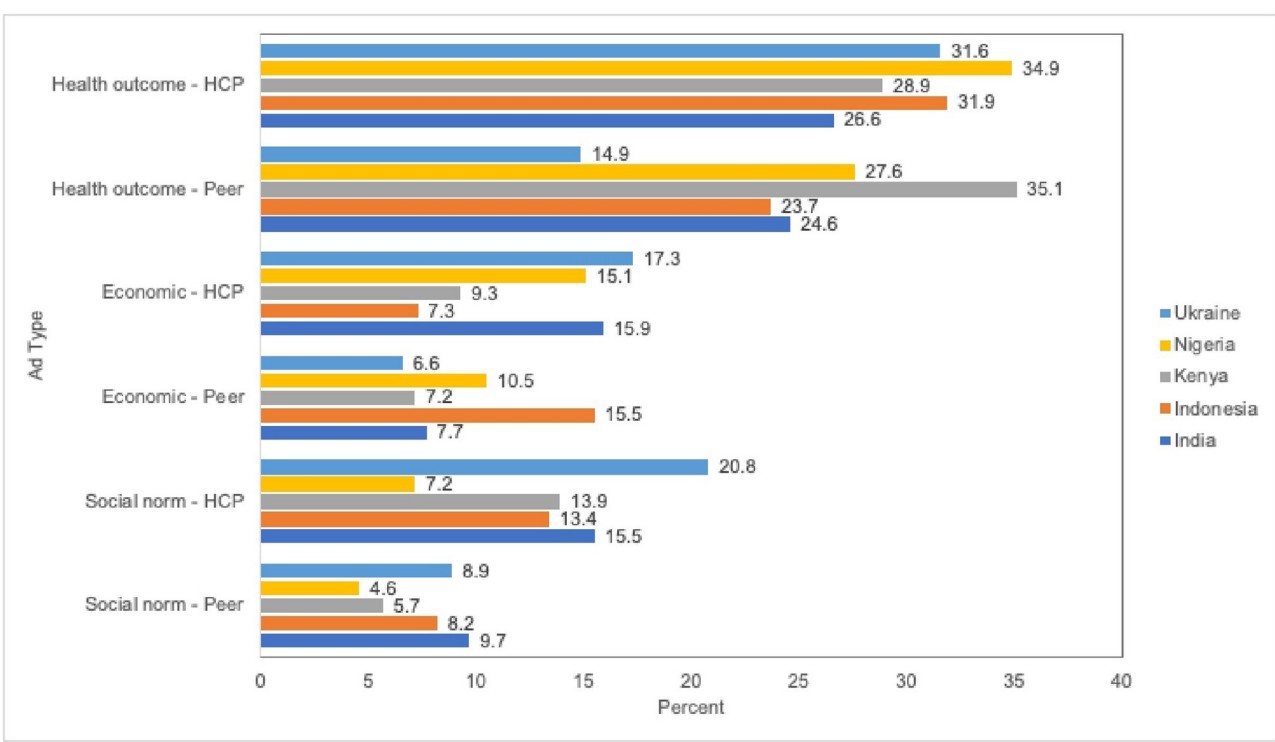

**Fig 2. Participants' preferred vaccine message among the six ads**\*. \* Pearson chi2(20) = 56.8347 Pr = 0.000.

indicated participants reported the health outcome–peer ad (92.3%) as the ad that would prompt them the most to tell someone about the COVID-19 vaccination. Participants found the health outcome–healthcare provider ad to be relevant to them (92.3%), motivated parents to vaccinate their children (67.5%), and motivated them to get the COVID-19 vaccine (91.9%).

Multivariable multinomial regression models were used to examine individual countries with ad preference. Indonesian participants with a graduate degree were more likely to prefer the economic outcome–healthcare provider ad (RR = 5.16, P = .05, 95% CI [1.03, 25.79]). Indonesian participants (RR = 6.46, P < .01, 95% CI [1.78, 23.52]) with higher vaccine hesitancy reported a preference for the social norm–healthcare provider ad (S2 Table). Male Ukrainian participants reported a preference social norm–peer ad (RR = 0.21, P = .03, 95% CI [.05, .87]) (S5 Table). Kenyan male participants (RR = -1.25, P = .04, 95% CI [-2.30, -0.20]) and participants with moderate/high vaccine hesitancy (RR = -1.21, P = .02, 95% CI [-2.37, -0.06]) were less likely to prefer the social norm–healthcare provider ad (S3 Table). Nigerian and Indian multivariable multinomial regression models did not indicate significant findings (S1 and S4 Tables).

We further examined the relationship between ad preference and pooled participant sociodemographic characteristics using multivariable multinomial regression (Table 3). Participants with a higher level of education were more likely to prefer the economic outcome-healthcare provider ad (RR = 3.20, P< .01, 95% CI [1.47, 6.97]). Male participants preferred the social norm-peer ad (RR = .56 P = .04, 95% CI [.0.32, 0.96]). Looking at country preferences, Ukrainian participants were more likely to prefer the health outcome-peer ad (RR = .49, P = .04, 95% CI [.25, .95]). Nigerian participants were more likely to prefer the social norm-healthcare provider ad (RR = .36, P = .02, 95% CI [.16, .82]). Nigerian participants also reported a preference for the social norm-peer ad (RR = .33, P = .03, 95% CI [.13, .88]).

**Table 3. Relative risk ratios of ad preference by vaccine hesitancy status and participant characteristics using multivariable multinomial logistic regression modeling (n = 900\*).**

| | Adjusted relative risk ratios (95% CI) | | | | |
|---|---|---|---|---|---|
| | **Health Outcome Peer** | **Economic Healthcare provider** | **Economic Peer** | **Social norm Healthcare provider** | **Social norm Peer** |
| **Country** | | | | | |
| India | Ref | Ref | Ref | Ref | Ref |
| Indonesia | 0.80 (0.46, 1.41) | 0.57 (0.27, 1.19) | 1.48 (0.70, 3.12) | 0.84 (0.44, 1.63) | 0.71 (0.32, 1.56) |
| Kenya | 1.18 (0.68, 2.07) | 0.56 (0.26, 1.19) | 0.45 (0.70, 3.11) | 0.80 (0.40, 1.60) | 0.42 (0.17, 1.05) |
| Nigeria | 0.80 (0.45, 1.44) | 0.77 (0.38, 1.56) | 0.85 (0.37, 1.94) | **0.36 (0.16, 0.82)** | **0.33 (0.13, 0.88)** |
| Ukraine | **0.49 (0.25, 0.95)** | 0.99 (0.49, 2.00) | 0.51 (0.20, 1.30) | 1.06 (0.53, 2.10) | 0.79 (0.34, 1.87) |
| **Vaccine hesitancy** | | | | | |
| Lower | Ref | Ref | Ref | Ref | Ref |
| Higher | 1.36 (0.90, 2.07) | 1.26 (0.74, 2.15) | 1.49 (0.83, 2.67) | 1.10 (0.68, 1.79) | 1.30 (0.70, 2.41) |
| **Age** | | | | | |
| <40 | Ref | Ref | Ref | Ref | Ref |
| 40+ | 0.64 (0.34, 1.21) | 0.64 (0.29, 1.41) | 1.46 (0.62, 3.35) | 1.13 (0.56, 2.27) | 0.65 (0.25, 1.71) |
| **Gender** | | | | | |
| Female | Ref | Ref | Ref | Ref | Ref |
| Male | 0.72 (0.50, 1.03) | 0.84 (0.53, 1.31) | 0.65 (0.40, 1.07) | 0.79 (0.52, 1.21) | **0.56 (0.32, 0.96)** |
| **Education** | | | | | |
| Secondary | Ref | Ref | Ref | Ref | Ref |
| Bachelor's Degree | 1.18 (0.74, 1.86) | 2.01 (.99, 4.01) | 0.83 (0.46, 1.53) | 1.08 (0.62, 1.90) | 1.20 (0.59, 2.42) |
| Graduate Degree | 1.30 (0.74, 2.29) | **3.20 (1.47, 6.97)** | 0.85 (0.39, 1.85) | 1.44 (0.75, 2.80) | 1.27 (0.54, 2.98) |

\* Reference category: health outcome / healthcare provider ad

We examined pooled participant ad messenger type (peer and healthcare provider) preference using logistic regression. Significant findings illustrate a preference for healthcare provider as the ad messenger. Male participants (RR = 1.33 P = .03, 95% CI [1.03, 1.73]) and participants aged 40+ years old (RR = 1.66 P = .03, 95% CI [1.06, 2.59]), were more likely to prefer healthcare providers as messengers. When looking specifically as the female pooled participant population, pregnant participants (RR = 1.75, P = .02, 95% CI [1.08, 2.83]) also were more likely to prefer healthcare providers as messengers. Pooled participant preference by ad type was also examined (health outcome, economic outcome, social norm) using a multinomial multivariable regression model. Using the health outcome ad type as the reference group, results indicated Kenyan participants were more likely to prefer the economic outcome ads (RR = .47, P = .01, 95% CI [.26, .85]) and Nigerian participants were more likely to prefer the social norm ads (RR = .39, P < 0.01, 95% CI [.21, .73]).

We also examined the relationship between ad choice and characteristics specifically in female-identified participants. Female participants with higher education were more likely to prefer the economic outcome-healthcare provider ad: bachelor's degree (RR = 5.61, P = .03, 95% CI [1.21, 25.99]) and graduate degree (RR = 9.67, P = .001, 95% CI [1.90, 49.16]). Female participants indicating they were currently pregnant were more likely to prefer the health outcome-peer ad (RR = .36, P = .01, 95% CI [.17, .77]). Like the overall Nigerian findings, female Nigerian participants were more likely to prefer the social norm-peer ad (RR = .14, P = .02, 95% CI [.27, .75]. Finally, among pregnant participants, pregnant participants from Kenya (OR = 2.6, 95% CI [1.26, 5.31]), Nigeria OR = 10.74, 95% CI [3.54, 32.61]), and Ukraine (OR = 2.6, 95% CI [1.26, 5.42]) were more likely to have higher hesitancy (Table 4).

**Table 4. Odds ratios of vaccine hesitancy status by pregnant participant characteristics using logistic regression modeling (n = 441)***.

| | Odds Ratio (95% CI) |
|---|---|
| **Country** | |
| India | Ref |
| Indonesia | 1.33 (0.70, 2.50) |
| Kenya | **2.59 (1.26, 5.31)** |
| Nigeria | **10.74 (3.54, 32.61)** |
| Ukraine | **2.61 (1.26, 5.42)** |
| **Age** | |
| <40 | Ref |
| 40+ | 0.93 (0.42, 2.03) |
| **Education** | |
| Secondary | Ref |
| Bachelor's Degree | 1.72 (0.98, 3.04) |
| Graduate Degree | 1.59 (0.78, 3.24) |
| **Currently Pregnant** | |
| No | Ref |
| Yes | 1.33 (0.74, 2.37) |

* Reference category: Lower vaccine hesitancy

## Discussion

As vaccination, not vaccines, saves lives, understanding how to improve persuasive communication to promote vaccine uptake is crucial. While both the message itself and the message appeal is critical to nudge individuals toward vaccination [18], the messenger also plays a key role in building trust in vaccines [19]. This is one of the first studies to examine and compare multi-country vaccine ad preferences in an online, English-speaking, adult population across a diverse set of low- and middle-income countries encompassing Asia, Africa, and Europe. This study has several key implications for informing persuasive messaging to improve vaccine uptake and provides important guidance for effectively engaging with distinct populations.

First, across all 5 countries, almost one-third of participants reported having ever delayed or refused a recommended vaccine. There are clearly concerns about vaccines, and more so among vaccines for pregnant women and children, regardless of country, which is not new [20]. This finding indicates a need to continue to identify specific concerns across populations to develop and implement approaches to overcome such concerns [21, 22]. Our results also point to the fact that across countries many people have concerns over the effectiveness of COVID-19 vaccines in preventing illness. When COVID-19 vaccine rollout began, there was a missed opportunity to frame these vaccines as a means in which to prevent *severe disease* or hospitalization rather than prevention of illness altogether. With refined messaging, these perceptions and concerns can be addressed. Correcting this misperception will also help manage expectations related to vaccine effectiveness [23].

Second, all six ads–all three appeals and both messengers–were received favorably across all five countries. These findings suggest that appeals related to health outcome, economic benefit, and social norms are all acceptable to diverse general populations, while specific audience segments (i.e., mothers, younger adults, etc.) may have preferences for certain appeals over others [24]. Our findings also showed that both health care providers and peers served as acceptable messengers. Given the decline in trust in health care systems globally [25], and hesitancy

among health care providers themselves [26], it is prudent to identify a variety of trusted messengers, including non-health care messengers, in order to effectively promote vaccination. Understanding the role of non-health care messengers has been studied in the US [27, 28], and studies have identified that faith-based leaders [29], for example, are credible and trusted messengers that could play crucial roles in promoting vaccination. Along with identifying additional trusted messengers, additional appeals for vaccination should also be tested in these countries to further aid effective audience segmentation [30, 31].

While this work identifies promising communication strategies that appeal to diverse, multi-national audiences, it also identifies country-specific differences that may aid the effectiveness of immunization appeals on a national level in LMIC settings. Regarding ad preference by country, Ukrainian participants were more likely to prefer the ad with a peer messenger and a health outcome appeal. Studies have shown that Ukraine is plagued with misinformation [32] and that there is low trust in the Ukrainian government [13], which may explain why a peer is preferred over a health provider as a persuasive messenger for vaccination. Nigerian participants were more likely to prefer appeals related to social norms with both healthcare provider and peer messengers. This demonstrates that social norms may be powerful influences for improving vaccine uptake in Nigeria, suggesting that identifying individuals that are perceived as peers, rather than hierarchical, may be useful for vaccine promotion [33].

Encouraging pregnant individuals to accept COVID-19 vaccines where they are available and recommended has been challenging globally. As most COVID-19 trials did not include pregnant participants [34], conveying the safety and effectiveness of COVID-19 vaccines during pregnancy continues to be a challenge. Pregnant individuals across all 5 countries tended to prefer the peer messenger. This finding is interesting, given the literature that suggests that pregnant women rely on their health care provider for vaccine recommendations to inform their own vaccine behavior [35]. It would be prudent to test additional peer messengers as potential channels for promoting vaccination among this group [36].

This study has several limitations. First, we relied on an online panel to test our message appeals and messengers. Selection bias associated with online surveys has been well documented [37], as such approaches tend to underrepresent individuals who are older, lacking internet access, have lower income, and have less education. We did not design this study to be a population representative sample. We were interested in understanding ad preferences using in this descriptive study, using samples from diverse countries. Although online surveys have several limitations as a research tool, they also provide a promising tool for researchers to reach diverse audiences outside of the more traditionally accessible WEIRD populations (Western, Educated, Industrialized, Rich, and Democratic) [38]. While these tools are not perfect, they do allow us to reach populations that are traditionally under-reached. This study's cross-sectional nature allows us to capture a moment in time, even though vaccine attitudes shift over time, given a multitude of factors.

Despite these limitations, this study is one of the first to test appeals and messengers in across a variety of low- and middle-income countries. To increase vaccine acceptance, identifying preferences for appeals and messengers is paramount. Tailored approaches, or approaches that are developed with the target audience's concerns and preferences in mind, will be more effective than broad-based or mass appeals [39]. Vaccine behavior is complex with many determinants. To ensure global vaccine uptake remains adequate, it is important to meet people where they are and to respond to their concerns through trusted messengers relevant to specific audiences and appeals that are salient and relevant. We are hopeful that this work will help inform future messaging and will inspire researchers and practitioners across the globe to examine how they can more effectively promote vaccines given their target audiences.

## Supporting information

**S1 Table. India relative risk ratios of ad preference by vaccine hesitancy status and participant characteristics using multivariable multinomial logistic regression modeling.**
(DOCX)

**S2 Table. Indonesia relative risk ratios of ad preference by vaccine hesitancy status and participant characteristics using multivariable multinomial logistic regression modeling.**
(DOCX)

**S3 Table. Kenya relative risk ratios of ad preference by vaccine hesitancy status and participant characteristics using multivariable multinomial logistic regression modeling.**
(DOCX)

**S4 Table. Nigeria relative risk ratios of ad preference by vaccine hesitancy status and participant characteristics using multivariable multinomial logistic regression modeling.**
(DOCX)

**S5 Table. Ukraine relative risk ratios of ad preference by vaccine hesitancy status and participant characteristics using multivariable multinomial logistic regression modeling.**
(DOCX)

## Author Contributions

**Conceptualization:** Rupali J. Limaye, Kristian Balgobin, Alexandra Michel, Gretchen Schulz, Daniel J. Erchick.

**Data curation:** Rupali J. Limaye, Kristian Balgobin, Alexandra Michel, Daniel J. Erchick.

**Formal analysis:** Rupali J. Limaye, Kristian Balgobin, Daniel J. Erchick.

**Funding acquisition:** Rupali J. Limaye, Alexandra Michel.

**Investigation:** Rupali J. Limaye, Kristian Balgobin, Alexandra Michel, Gretchen Schulz.

**Methodology:** Rupali J. Limaye, Kristian Balgobin, Alexandra Michel, Daniel J. Erchick.

**Project administration:** Rupali J. Limaye, Alexandra Michel, Gretchen Schulz.

**Resources:** Rupali J. Limaye, Alexandra Michel.

**Software:** Rupali J. Limaye, Alexandra Michel.

**Supervision:** Rupali J. Limaye, Alexandra Michel, Daniel J. Erchick.

**Validation:** Rupali J. Limaye, Daniel J. Erchick.

**Visualization:** Rupali J. Limaye, Alexandra Michel, Daniel J. Erchick.

**Writing – original draft:** Rupali J. Limaye, Kristian Balgobin, Alexandra Michel, Gretchen Schulz.

**Writing – review & editing:** Rupali J. Limaye, Kristian Balgobin, Alexandra Michel, Gretchen Schulz, Daniel J. Erchick.

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
