## [Decision Letter · Decision Letter 0]

3 Aug 2022

PONE-D-22-18465What message appeal and messenger are most persuasive for COVID-19 vaccine uptake: Results from a 5-country survey in India, Indonesia, Kenya, Nigeria, and UkrainePLOS ONE

Dear Dr. Limaye,

Thank you for submitting your manuscript to PLOS ONE. After careful consideration, we feel that it has merit but does not fully meet PLOS ONE’s publication criteria as it currently stands. Therefore, we invite you to submit a revised version of the manuscript that addresses the points raised during the review process.

We look forward to receiving your revised manuscript.

Kind regards,

Bassey E. Ebenso, Ph.D., M.P.H., M.D.,

Academic Editor

PLOS ONE

Journal Requirements:

a) Did participants provide their written or verbal informed consent to participate in this study?

4. We note you have included a table to which you do not refer in the text of your manuscript. Please ensure that you refer to Tables 5-9 in your text; if accepted, production will need this reference to link the reader to the Table.

5. Please include your tables as part of your main manuscript and remove the individual files. Please note that supplementary tables (should remain/ be uploaded) as separate "supporting information" files.

Additional Editor Comments:

Please use the comments provided by Reviewer 2 to revise and re-submit your mansucript.

Reviewers' comments:

Reviewer's Responses to Questions

**Comments to the Author**

1. Is the manuscript technically sound, and do the data support the conclusions?

Reviewer #1: Yes

Reviewer #2: Partly

2. Has the statistical analysis been performed appropriately and rigorously? 

Reviewer #1: Yes

Reviewer #2: Yes

3. Have the authors made all data underlying the findings in their manuscript fully available?

Reviewer #1: Yes

Reviewer #2: Yes

4. Is the manuscript presented in an intelligible fashion and written in standard English?

Reviewer #1: Yes

Reviewer #2: Yes

5. Review Comments to the Author

Reviewer #1: Abstract and Introduction

The abstract and introduction describes the importance of the study appropriately.

Methods

I think the method of data collection and analysis are appropriate.

Results, discussion, conclusions

• The discussions are write written appropriately.

•The conclusion should be a concise summery of the results.

Reviewer #2: The importance and timing of this work is excellent given the global disease challenges and the need for holistic preventive measures globally. The authors did a wonderful effort in conceptualizing and actualizing this work and the outcome of their analyses will be of huge significance if adopted by many countries.

Above notwithstanding, there are some fundamental observations which when addressed will surely improve the quality and acceptability of the outcome and recommendations from this work. These observations are listed below but were not in any way arranged based on importance.

1. There was no mention of any pre-testing of the survey instrument to ascertain its validity and reliability.

2. There was no indication of how minimum sample size for this study was computed. This is very important when the populations of the countries were considered. Combined, these countries have a total of over 1.6 billion people, and to analyze responses from ‘just’ 953 respondents may not guarantee the projected outcomes if the conclusions/recommendations from this work were to be adopted by the concerned countries.

3. With about 48.6% (n = 921) of the responses rejected due to failure to pass quality check, it is obvious that the medium employed to conduct this survey is not very reliable. As the authors set out to assess (in part) message appeal, they ought to have also assess many online survey tools to be able to see which one will give them the highest quality-check-adherent responses.

4. There are some inconsistences in how some figures were computed, some specific examples are:

a. Percentages were correctly in line 201: based on age ( 463÷953; 48.6%) and based on female gender ( 487÷950; 52.3%) but the denominator with which the percentage in line 202 (based on Bachelor’s degree) was computed was not very clear.

b. Sum total of respondents are not consistent across some variables in Table 1. Example, while the totals for country, age, gender and vaccine hesitancy are 953 each, the total for education is 901 (not 953), that for females (based on pregnancy status) is 399 (instead of 487) and for COVID-19 vaccination is 930 instead of 953.

5. The authors should highlight the difference between incomplete surveys (n=20; Figure 1) that were excluded from analysis, and those termed ‘missingness’ as captured on the footnote attached to Table 1. This is pertinent because with significant number of missing responses (e.g. education, n = 52, and COVID-19 vaccinated, n = 23) the validity of the whole analysis can be questioned. That is to say, if initially 20 responses were rejected for incomplete survey (giving rise to 953 total valid responses) and now another 52 (based on education level) again disqualified, further dropping the total valid responses to 901, it cast doubt on the validity of the logistic regression analysis conducted.

Conclusion:

1. If the sample size used for this work is proven (by the authors) to have been arrived at using accepted scientific methods, the I will recommend that the article be accepted with minor corrections, such as computational errors, e.t.c.

2. If the sample size used cannot be proven (by the authors) to be scientifically adequate for inference to be drawn, then I will recommend that the article be rejected due to insufficient data, BUT it should be recommended that the authors should increase the sample size viz a viz data volume, re-do the analyses and resubmit the article.

6. PLOS authors have the option to publish the peer review history of their article (what does this mean?). If published, this will include your full peer review and any attached files.

Reviewer #1: **Yes: **Mohammad Aminul Islam

Reviewer #2: **Yes: **Aliyu Musawa Ibrahim

---

## [Author Response · Author response to Decision Letter 0]

24 Aug 2022

Response to Reviewer Comments 

1. Please ensure that your manuscript meets PLOS ONE's style requirements, including those for file naming. The PLOS ONE style templates can be found at  

https://journals.plos.org/plosone/s/file?id=wjVg/PLOSOne_formatting_sample_main_body.pdf and  

Manuscript format has been updated to meet PLOS ONE style requirements. 

a) Did participants provide their written or verbal informed consent to participate in this study? 

We have included this information in the methods section. A brief consent statement appeared on the screen used for the survey and informed consent was obtained online through the presentation of the consent message. 

Figures have been updated to meet PLOS ONE style requirements. 

4. We note you have included a table to which you do not refer in the text of your manuscript. Please ensure that you refer to Tables 5-9 in your text; if accepted, production will need this reference to link the reader to the Table. 

Tables previously referred to as Tables 5-9 have been renamed to match PLOS format as supporting information tables. S1-S5 Table will be submitted as supporting information and their titles have been added to the end of the manuscript per PLOS ONE style guidelines. 

5. Please include your tables as part of your main manuscript and remove the individual files. Please note that supplementary tables (should remain/ be uploaded) as separate "supporting information" files. 

Tables have been updated to meet PLOS ONE style requirements 

Confirming the reference list and references are accurate. 

Additional Editor Comments: 

Please use the comments provided by Reviewer 2 to revise and re-submit your mansucript. 

Reviewers' comments: 

Reviewer #1: Abstract and Introduction 

The abstract and introduction describes the importance of the study appropriately. 

Thank you for this comment. 

Methods 

I think the method of data collection and analysis are appropriate. 

Thank you for this comment. 

Results, discussion, conclusions 

• The discussions are write written appropriately. 

•The conclusion should be a concise summery of the results. 

Thank you for these comments. 

Reviewer #2: The importance and timing of this work is excellent given the global disease challenges and the need for holistic preventive measures globally. The authors did a wonderful effort in conceptualizing and actualizing this work and the outcome of their analyses will be of huge significance if adopted by many countries. 

Thank you for these comments. 

Above notwithstanding, there are some fundamental observations which when addressed will surely improve the quality and acceptability of the outcome and recommendations from this work. These observations are listed below but were not in any way arranged based on importance. 

Thank you for these comments. We appreciate your review. 

1. There was no mention of any pre-testing of the survey instrument to ascertain its validity and reliability. 

Thank you for flagging this. We did pre-test the survey and we have included information related to this manuscript. 

2. There was no indication of how minimum sample size for this study was computed. This is very important when the populations of the countries were considered. Combined, these countries have a total of over 1.6 billion people, and to analyze responses from ‘just’ 953 respondents may not guarantee the projected outcomes if the conclusions/recommendations from this work were to be adopted by the concerned countries. 

Thank you for this comment. This was an exploratory study, and as such, we did not have aprori hypotheses that we were testing. We did not design this to be a representative population study. We have included this information in the methods and limitations. 

3. With about 48.6% (n = 921) of the responses rejected due to failure to pass quality check, it is obvious that the medium employed to conduct this survey is not very reliable. As the authors set out to assess (in part) message appeal, they ought to have also assess many online survey tools to be able to see which one will give them the highest quality-check-adherent responses. 

We have stated limitations related to online survey research in the methods and limitations, including the use of the platform we used. We sought to use quality checks to ensure that answers we included in our analysis were of high quality. 

4. There are some inconsistences in how some figures were computed, some specific examples are: 

a. Percentages were correctly in line 201: based on age ( 463÷953; 48.6%) and based on female gender ( 487÷950; 52.3%) but the denominator with which the percentage in line 202 (based on Bachelor’s degree) was computed was not very clear. 

For demographics questions, participants were coding as missing when providing the response “Other.” 

b. Sum total of respondents are not consistent across some variables in Table 1. Example, while the totals for country, age, gender and vaccine hesitancy are 953 each, the total for education is 901 (not 953), that for females (based on pregnancy status) is 399 (instead of 487) and for COVID-19 vaccination is 930 instead of 953. 

For demographics questions, participants were coding as missing when providing the response “Other.” For pregnancy status, this was only examined among female identified participants, there were four categories (Female, Male, Non-Binary, and Other). 

5. The authors should highlight the difference between incomplete surveys (n=20; Figure 1) that were excluded from analysis, and those termed ‘missingness’ as captured on the footnote attached to Table 1. This is pertinent because with significant number of missing responses (e.g. education, n = 52, and COVID-19 vaccinated, n = 23) the validity of the whole analysis can be questioned. That is to say, if initially 20 responses were rejected for incomplete survey (giving rise to 953 total valid responses) and now another 52 (based on education level) again disqualified, further dropping the total valid responses to 901, it cast doubt on the validity of the logistic regression analysis conducted. 

While participants completed the survey in its entirety, for specific questions, responses were coded as missing due to their low frequency. For demographics questions, participants were coding as missing when providing the response “Other.” For questions related to vaccine hesitancy, participant responses “Don’t know” were also coded as missing. 

Conclusion: 

1. If the sample size used for this work is proven (by the authors) to have been arrived at using accepted scientific methods, the I will recommend that the article be accepted with minor corrections, such as computational errors, e.t.c. 

Thank you for your suggestion. We have included a sample size calculation in the methods. 

2. If the sample size used cannot be proven (by the authors) to be scientifically adequate for inference to be drawn, then I will recommend that the article be rejected due to insufficient data, BUT it should be recommended that the authors should increase the sample size viz a viz data volume, re-do the analyses and resubmit the article. 

Thank you for your suggestion. We have included a sample size calculation in the methods.

---

## [Decision Letter · Decision Letter 1]

8 Sep 2022

What message appeal and messenger are most persuasive for COVID-19 vaccine uptake: Results from a 5-country survey in India, Indonesia, Kenya, Nigeria, and Ukraine

PONE-D-22-18465R1

Dear Dr. Limaye,

We’re pleased to inform you that your manuscript has been judged scientifically suitable for publication and will be formally accepted for publication once it meets all outstanding technical requirements.

Kind regards,

Bassey E. Ebenso, Ph.D., M.P.H., M.D.,

Academic Editor

PLOS ONE

Additional Editor Comments (optional):

Your revised manuscript (Revision number 1) sufficiently addressed all comments raised by reviewer 2.

Reviewers' comments:

Reviewer's Responses to Questions

**Comments to the Author**

1. If the authors have adequately addressed your comments raised in a previous round of review and you feel that this manuscript is now acceptable for publication, you may indicate that here to bypass the “Comments to the Author” section, enter your conflict of interest statement in the “Confidential to Editor” section, and submit your "Accept" recommendation.

Reviewer #2: All comments have been addressed

2. Is the manuscript technically sound, and do the data support the conclusions?

Reviewer #2: Yes

3. Has the statistical analysis been performed appropriately and rigorously? 

Reviewer #2: Yes

4. Have the authors made all data underlying the findings in their manuscript fully available?

Reviewer #2: Yes

5. Is the manuscript presented in an intelligible fashion and written in standard English?

Reviewer #2: Yes

6. Review Comments to the Author

Reviewer #2: Authors should clearly indicate where modifications were made during data coding and entry, this is to avoid confusing the reader. Such modifications occurred during coding demographic data.

7. PLOS authors have the option to publish the peer review history of their article (what does this mean?). If published, this will include your full peer review and any attached files.

Reviewer #2: **Yes: **Aliyu Musawa Ibrahim

---

## [Editor Report · Acceptance letter]

12 Sep 2022

PONE-D-22-18465R1 

What message appeal and messenger are most persuasive for COVID-19 vaccine uptake: Results from a 5-country survey in India, Indonesia, Kenya, Nigeria, and Ukraine 

Dear Dr. Limaye:

I'm pleased to inform you that your manuscript has been deemed suitable for publication in PLOS ONE. Congratulations! Your manuscript is now with our production department. 

Kind regards, 

on behalf of

Dr. Bassey E. Ebenso 

Academic Editor

PLOS ONE